# YouTube-ASL: A Large-Scale, Open-Domain American Sign Language-English Parallel Corpus

**David Uthus, Garrett Tanzer, Manfred Georg**
Google
{duthus,gtanzer,mgeorg}@google.com

## Abstract

Machine learning for sign languages is bottlenecked by data. In this paper, we present YouTube-ASL, a large-scale, open-domain corpus of American Sign Language (ASL) videos and accompanying English captions drawn from YouTube. With ~1000 hours of videos and >2500 unique signers, YouTube-ASL is ~3x as large and has ~10x as many unique signers as the largest prior ASL dataset. We train baseline models for ASL to English translation on YouTube-ASL and evaluate them on How2Sign, where we achieve a new finetuned state of the art of 12.39 BLEU and, for the first time, report zero-shot results.

## 1 Introduction

The primary bottleneck for machine learning research on sign languages is data. As minority languages used by historically marginalized Deaf/Hard of Hearing communities, sign languages lack the plentiful online resources that have facilitated modern machine learning advances [4, 44, 12]. This is compounded by the fact that sign languages have no standardized written form: mining the videos that do exist is more difficult than retrieval for spoken language text. For translation specifically, there is the added problem of finding spoken language captions that are aligned to corresponding sign language content, rather than a voiceover with its own timing. The result is that datasets tend to be constructed by recording new footage in a studio or curating videos from a small number of manually selected content creators, which limits variety.

In order to address these challenges, we present YouTube-ASL, a large-scale, open-domain corpus of American Sign Language (ASL) videos and accompanying English captions, primarily intended for ASL to English machine translation. We mined these videos from YouTube using a two-step process: first, we used automatic content-based annotations to identify potentially relevant captioned videos; and second, we used skilled human annotators to filter out videos with poor quality or misaligned captions. The result is a dataset with 984 hours of high-quality captioned video featuring >2500 unique signers, which is ~3x as large as the largest prior ASL dataset [39] and has ~10x as many unique signers as any sign language dataset to date.

We train simple baseline models for sentence-level ASL to English translation on YouTube-ASL by embedding MediaPipe Holistic landmarks [26, 16] into the T5 language model [33]. Because YouTube videos may be removed over time and therefore cannot form a stable test set—and for comparison to prior work—we evaluate on a standard benchmark, How2Sign [13]. Borrowing from trends in mainstream machine learning [32, 15, 10], we provide not just finetuned but also zero-shot[1] results to test out-of-domain generalization. We achieve a new finetuned state of the art of 12.39 BLEU (vs. the prior SOTA of 8.03 [40]), and for the first time report a zero-shot score, 3.95 BLEU.

---

[1]Here, "zero-shot" refers to evaluation on an independently constructed benchmark without any kind of domain adaptation. This is different from the use of "zero-shot" in machine translation for transfer to unseen language pairs, or "zero-shot" in prompting for prompts without in-context examples.

37th Conference on Neural Information Processing Systems (NeurIPS 2023) Track on Datasets and Benchmarks.

Table 1: Summary statistics for different sign language translation datasets. See Section 3.3 for details on how these statistics were derived for YouTube-ASL.

| Name | Language | Vocab. | # Hours | # Signers | Source |
|---|---|---|---|---|---|
| RWTH-PHOENIX-2014T [5] | DGS | 3K | 11 | 9 | TV |
| BOBSL [2] | BSL | 77K | 1447 | 39 | TV |
| SWISSTXT [7] | DSGS | - | 88 | - | TV |
| VRT-RAW [7] | VGT | - | 100 | - | TV |
| CSL-Daily [47] | CSL | 2K | 23 | 10 | Lab |
| KETI [22] | KVK | 419 | 28 | 14 | Lab |
| Public DGS Corpus [18] | DGS | - | 50 | - | Lab |
| SP-10 [42] | various | 17K | 14 | 79 | Web |
| AfriSign [17] | various | 20K | 152 | - | Web |
| How2Sign [13] | ASL | 16K | 79 | 11 | Lab |
| OpenASL [39] | ASL | 33K | 288 | 220 | Web |
| YouTube-ASL (ours) | ASL | 60K | 984 | >2519 | Web |

We publicly release the YouTube-ASL video IDs.[2] We hope that YouTube-ASL will be useful for general sign language pretraining, as well as downstream tasks such as ASL to English translation and caption alignment—both in the near term to aid in the construction of larger sign language datasets, and eventually to improve accessibility for the Deaf/Hard of Hearing community.

## 2 Related Work

In this section, we review prior sign language translation datasets and methods for translation from sign languages to spoken languages.

### 2.1 Sign Language Translation Datasets

Table 1 shows statistics on different sign language translation datasets. There are three main sources for sign language data: ad hoc recorded footage, interpreted TV broadcasts, and online video sharing platforms.

In the first category are datasets that manually recruit signers and record them performing translations of desired phrases, either in a lab setting or with a camera on their personal device. These datasets tend to be small and feature few signers for logistical reasons, and may have exhaustive annotations because the small size of the dataset makes it feasible. This includes datasets such as CSL-Daily [47], with phrases related to daily life in Chinese Sign Language; KETI [22], with phrases related to emergency situations in Korean Sign Language; Public DGS Corpus [18], with elicited dialogues in German Sign Language; and How2Sign [13], with "How To" instructional monologues translated into American Sign Language.

In the second category are datasets that collate interpreted TV programs from a collaborating national broadcaster. These datasets tend to be larger than newly recorded ones, but often use a small number of non-native interpreters and lack fine-grained caption alignment (because the supervision comes from the spoken language audio track). This includes datasets such as RWTH-PHOENIX-2014 [5], with weather forecasts interpreted into German Sign Language; SWISSTXT [7], with news/weather programs interpreted into Swiss German Sign Language; VRT [7], with news programs interpreted into Flemish Sign Language; and BOBSL [2], with BBC programs in many domains interpreted into British Sign Language. At 1447 hours, BOBSL is the largest sign language translation dataset to date (including the present work), but has only 39 signers and speech-aligned subtitles, vs. YouTube-ASL's >2519 signers and sign-aligned captions—though the two datasets are complementary because they are for different languages.

In the third category are datasets that curate content from online video sharing platforms. In prior sign language translation datasets, this content is drawn from a small number of manually selected channels.

---

[2]https://github.com/google-research/google-research/tree/master/youtube_asl

This includes datasets such as SP-10 [42], with example sentences from an online multilingual sign dictionary; AfriSign [17], with translated Bible passages hosted on the Jehovah's Witnesses website; and OpenASL [39], with videos from three YouTube channels: *DailyMoth*, *Sign1News*, and the National Association of the Deaf. OpenASL is the largest prior ASL dataset and closest work to YouTube-ASL: the key difference is that YouTube-ASL is constructed with open-ended mining from automatic tags, rather than manual channel curation. OpenASL is largely a subset of YouTube-ASL, which—by utilizing the long tail of channels—is ~3x as large and has ~10x as many unique signers.

There are several datasets for easier tasks than translation, like isolated sign recognition and finger-spelling recognition, that mine from the web by ambiguous means. MS-ASL [21], WLASL [24], ChicagoFSWild [37]/ChicagoFSWild+ [38], CISLR [19], and Indian-SL [35] are word-level datasets mined from YouTube, sign language-targeted sites like ASLU and ASL-LEX, or other unnamed video sharing platforms. These works do not specify how they retrieved their videos, so it is possible that they used a similar automatic tagging approach to YouTube-ASL, albeit on a more limited scale.

## 2.2 End-to-End Sign Language Translation

Originally, sign language translation approaches operated on *glosses*, linguistic annotations that represent individual signs, or cascaded translation through glosses as an intermediate step, like speech to text translation often cascades through speech recognition. More recently, due to a variety of deficiencies in glosses and lack of widespread gloss data, the field has shifted to end-to-end modeling with encoder-decoder Transformers, starting with Camgöz et al. [5].

The two main classes of approaches are those that take learned video embeddings as input [6, 40, 28] (via video encoders, primarily I3D [9], pretrained on tasks such as isolated sign recognition), and those that take estimated pose landmarks as input [28] (such as MediaPipe [26] or OpenPose [8]). Some works achieve modest gains given constant data with architectural tweaks like treating different cues in the input video (hands, face) differently [46, 43]. It is unclear to what extent these techniques are necessary or beneficial on larger datasets. Other works seek to benefit from transfer from spoken language or other sign language data [11, 45, 17]. All of these works train and evaluate on splits derived from the same underlying continuous sign language corpus (different datasets across papers), and sometimes multiple such datasets independently in the same paper. In contrast, we train on YouTube-ASL using an uncomplicated approach and evaluate on How2Sign, reporting both finetuned and zero-shot results to get a more robust understanding of our model's state-of-the-art performance.

## 3 The YouTube-ASL Corpus

YouTube-ASL is a corpus of American Sign Language (ASL) videos with accompanying English captions drawn from YouTube. Video sharing platforms like YouTube are appealing sources of sign language data because they host swaths of diverse content that are more broadly representative of real world conditions than studio footage is. Of course, much of this data is irrelevant or low-quality, so it is imperative to develop cost-effective ways to sift through it.

We used a two-step pipeline to construct the corpus: first, retrieval using automatic content-based annotations, and second, filtering by skilled human annotators at a per-video level. This automatic retrieval step represents a departure from prior continuous sign language corpora and brings us closer to mining approaches from mainstream machine learning.

### 3.1 Automatically Retrieving Candidate Videos

As described previously in Abu-El-Haija et al. [1], the YouTube video annotation system associates machine-generated tags with each video in the form of Knowledge Graph entities, which are based on the video's metadata, context, and content signals. We retrieved listed public videos tagged as being related to sign language generally or American Sign Language specifically, as of January 2022.[3] This automatic tagging step, while having higher recall than prior works, was flawed in that it was not aware of sign language in the video content itself—to be expected due to the limited nature of current sign language processing. This means that, for example, videos in sign language that do not explicitly mention sign language in the content or context were unlikely to be discovered. This failure

---

[3]Some video IDs may have been removed from this set over time due to video deletions.

mode was most salient for press conferences with simultaneous interpreters, which tend not to have well-aligned captions anyway.

Given these retrieved videos, we drilled down on those with user-generated captions—i.e., captions that were manually uploaded rather than automatically derived from speech—because speech-derived captions are not tightly aligned with signed content. As a heuristic filtering step, we automatically removed videos with duration <10 seconds or >5 hours, width <480 pixels or height >360 pixels, and frame rate <15fps or >60fps. We arrived at these values through an iterative mining and auditing process, so that we could reduce annotator labor spent on irrelevant videos without excluding too many relevant ones. From inspection, the heuristic excluded a negligible amount of desirable videos. The one class of useful videos one might expect this to exclude, short isolated sign videos as used by MS-ASL [21] and WLASL [24], tends to have the label in the video title or description rather than captions, so removing videos under 10 seconds does not have a substantial impact. Videos that were over 5 hours were often either live interpreted broadcasts (which did not have aligned captions) or not sign language (e.g., corrupted videos or mostly static content for hours on end).

Finally, we used off-the-shelf person detection tools to exclude videos where none of the captions corresponded to spans with exactly one person present in the video. We limit the scope of our efforts to signing monologues due to the challenges of modeling conversations between multiple signers.

The result was a list of 88,002 candidate videos that might contain ASL with high-quality captions.

## 3.2 Identifying High-Quality Videos with Skilled Human Annotators

While some smaller datasets like How2Sign [13] use annotators to manually align all captions, this becomes prohibitively expensive for larger datasets. For this reason, OpenASL [39] and BOBSL [2] use annotators to correct only their validation and test sets. We take a coarser-grained approach to annotations but apply it to our entire list of 88,002 candidates: we use humans to identify videos that are roughly suitable and include them in our corpus without modification.

To do so, we hired 3 native ASL signers with English proficiency to serve as annotators. The annotators used a bespoke internal tool that would display a given YouTube video and present label options. In order to save time, the annotators were able to mark that their labels held for an entire channel of videos rather than each video individually. Therefore it is possible that certain videos in the corpus are channel outliers and do not meet quality standards, but generally large channels have consistent quality. Each video was labelled by only one annotator unless they brought it up for wider discussion.

Through an iterative process involving written instructions, virtual meetings (through an ASL interpreter or signing project members), and escalations by email for edge cases, we aligned on standards for when to accept a video into the corpus. Some of the reasons for exclusion include: the video's captions do not exclusively correspond to signing; the video is in a sign language other than ASL; the video's captions do not correctly translate the ASL; and the captions are poorly aligned [4]. Notably, in order to increase the size of the corpus, we chose to include videos across all skill levels and signing styles, as long as they were comprehensible to an ASL user and correctly captioned. This variety is beneficial for sign language recognition tasks, where models should be able to understand all signers, but may limit the corpus's usefulness for generation tasks, where consistency and controllability are important.

The result was a list of 11,093 videos whose captions are generally well-aligned English translations of signed ASL content.

## 3.3 Corpus Statistics

The final, human-filtered YouTube-ASL corpus consists of 11,093 ASL videos with 984 total hours of footage. This is ~3x the size of OpenASL [39], the largest prior ASL dataset, but smaller than BOBSL [2], a British Sign Language dataset. See Table 1 for a comparison between the high-level

---

[4]For caption alignment, we are targeting captions that start and stop in close proximity to the signing, and the timings appear to have been annotated based on the video rather than a voice-over. Within a sentence, which may consist of multiple captions, there may be but is not necessarily an alignment because the syntax of ASL is different from English.

Table 2: Statistics on the distribution of captions and videos in the YouTube-ASL corpus.

| | |
|---|---|
| Number of captions | 610,193 |
| Caption length (Average / $90^{th}$ percentile, in characters) | 48.9 / 88.0 |
| Caption length (Average / $90^{th}$ percentile, in words) | 8.8 / 16.0 |
| Caption duration (Average / $90^{th}$ percentile, in seconds) | 4.8 / 8.76 |
| Video duration (Average / $90^{th}$ percentile, in seconds) | 318.95 / 675.80 |

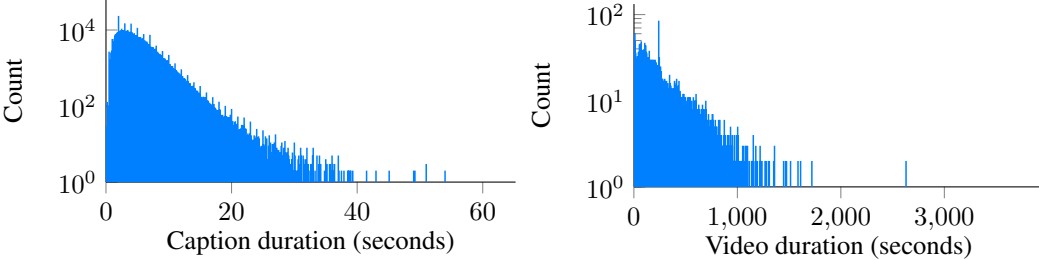

Figure 1: Distribution of caption and video durations. For the video duration graph, we omit 27 videos whose duration exceeds 3600 seconds (between 3610 and 9017 seconds).

attributes of YouTube-ASL and prior sign language translation datasets, including total number of hours.

These videos are paired with 610,193 English captions, with a total duration of 813 hours. See Table 2 for statistics on the distribution of captions, as well as Figure 1 for visualizations. The average caption length (8.8 words) and duration (4.8 seconds) are relatively short, which reflects that sentences may be split across multiple captions. We computed vocabulary size by counting the number of distinct strings between whitespace or punctuation across all captions. It is important to keep in mind that in addition to the signing itself, these videos' captions vary in style, literalness of translation (whether the content was originally produced in ASL and translated, or translated into ASL from these captions), spelling/grammar correctness, and more. This degree of variability is difficult to quantify in comparisons between datasets.

We use the number of unique channels, 2519, as an approximate lower bound for the number of unique signers in the dataset: some channels may feature many signers, and some signers may appear across multiple channels. Note that with this method, OpenASL [39] would be estimated to have 3 signers, while its authors reached a count of 220 signers using more fine-grained methods. Even this likely underestimate is ~10x the count of any individual sign language dataset to date.

Figure 2 shows the distribution of videos per channel, for channels with at least 20 videos. There are a few channels with many videos—in particular, the two largest channels are the same news channels featured in OpenASL—and then a long tail of channels with fewer videos. This means that the bulk of new footage present in YouTube-ASL but not OpenASL comes from relatively small channels, which helps variety. See Figure 3 for a sense of the distribution of (machine-annotated) topics across videos: they seem more diverse than prior datasets from video sharing platforms but still shaped by typical YouTube use cases, compared to BOBSL's more topic-balanced BBC programming.

## 4   Baseline Approach

In order to demonstrate the potential of YouTube-ASL, we consider a simple method for sentence-level machine translation from ASL to English built using off-the-shelf components. We use a deliberately barebones approach to avoid introducing inductive bias that helps in more limited settings but becomes harmful with scale.

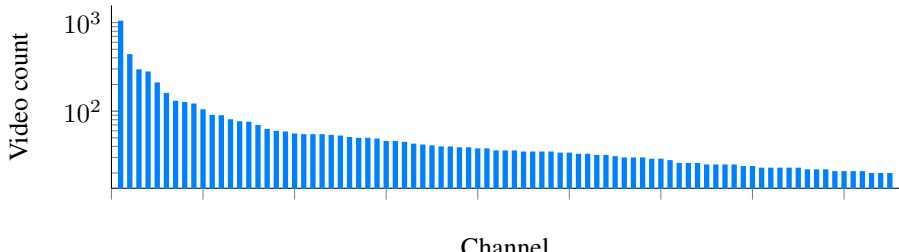

Figure 2: Distribution of videos per channel for channels with at least 20 videos.

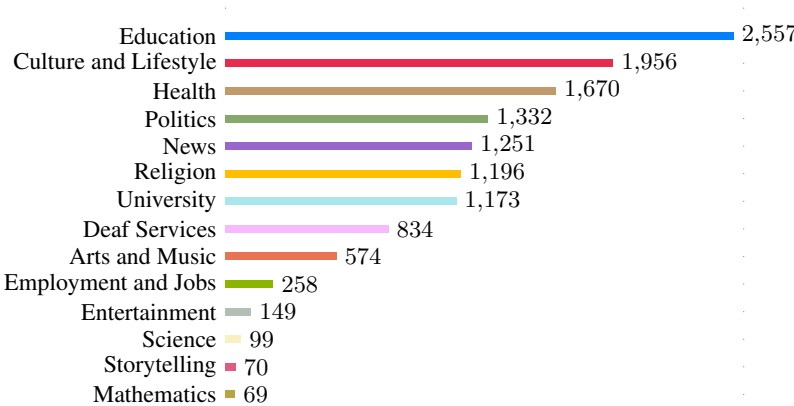

Figure 3: A selection of high-level topics, with the number of YouTube-ASL videos automatically tagged as related to them. Note that a single video can be tagged with more than one topic.

## 4.1 Preprocessing

For our target English outputs, we use the raw captions from YouTube-ASL. Each training example is clipped to the boundaries of a single caption. We filter out captions with length >300 characters or duration <200ms or >60s, which tend to be malformed, and any captions corresponding to video spans where exactly one person is not present. We do not lowercase the captions or apply any other kind of text normalization.

For our sign language inputs, we use MediaPipe Holistic landmarks [26, 16], rather than raw video. Sign language models that use pose-based inputs have a history of underperforming those that operate on learned video embeddings [21, 27]; it is unclear to what extent this is due to the information bottleneck in the (imperfectly predicted) pose representation, vs. availability of higher quality pretrained video encoders than pretrained pose encoders. Pose inputs offer some benefits like computational efficiency and privacy.

MediaPipe Holistic is a lightweight model that predicts 532 3D landmarks (in x-, y-, and z- image-space coordinates) for the hands, pose, and face of a single human in video footage. For sign language understanding tasks, many of these landmarks are redundant (high-detail face mesh) or unnecessary (lower body), and add undesirable complexity. We discard all but 85 of these points, selected *a priori* using domain knowledge about sign languages:

- For each hand, we use all 21 landmark points.

- For the pose, we use 6 landmark points, for the shoulders, elbows and hips.[5] This discards the lower body and pose landmarks redundant with the hand and face modules.

- For the face, we use 37 landmark points, from the eyes, eyebrows, lips, and face outline.[6]

---

[5]These are indices 11, 12, 13, 14, 23, 24.

[6]These are indices 0, 4, 13, 14, 17, 33, 37, 39, 46, 52, 55, 61, 64, 81, 82, 93, 133, 151, 152, 159, 172, 178, 181, 263, 269, 276, 282, 285, 291, 294, 311, 323, 362, 386, 397, 468, 473.

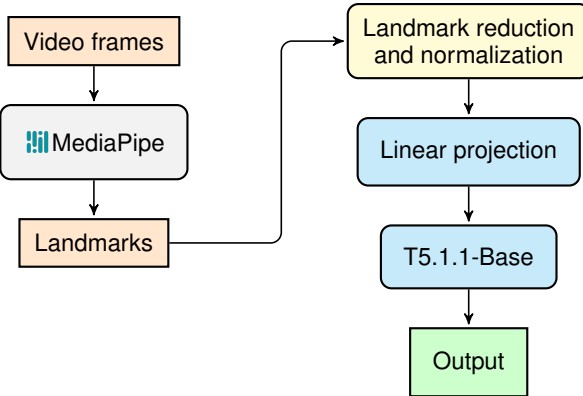

Figure 4: Overview of our model pipeline. Starting from an ASL video clip, we use MediaPipe Holistic to compute 3D landmarks for the face, hands, and body of the subject. We then discard irrelevant landmarks and normalize the remainder. These are concatenated and embedded by a linear projection layer into T5.1.1-Base, which then decodes the English translation. The blue components (Linear projection and T5) are the trainable parameters.

We normalize the landmarks by scaling them to fit in a unit bounding box across the duration of the clip. We represent landmarks that are not present in a frame with a large negative value. MediaPipe also predicts visibility (self-occlusion) of landmarks within the frame, which we ignore. To reduce sequence length, we discard every second frame. The final preprocessed input is therefore a half-frame rate sequence of 255-dimensional landmark vectors. Note that this half frame rate may vary from 7.5 to 30fps depending on the original video's frame rate, though most end up at 12 to 15 fps.

## 4.2  Model

Our model is a slightly modified version of T5 [33], which is an encoder-decoder Transformer [41] that has been trained on web-crawled English text. Rather than embed text tokens using a vocabulary of learned embeddings, we embed each 255-dimensional landmark frame into the encoder using a learned linear projection layer. Otherwise, our architecture is identical to T5.1.1-Base.

We set the encoder context window to 256 tokens (frames) and the decoder context window to 128 tokens, which accommodate the training examples after halving the input frame rate and encoding the target text with T5's SentencePiece vocabulary [23].

## 5  Experiments

We choose not to provide train, validation, and test splits for YouTube-ASL. Because YouTube videos may be deleted over time, the validation and test splits could not serve as a stable benchmark. We instead evaluate on How2Sign [13], a studio-recorded dataset released under CC BY-NC 4.0 consisting of "How To" instructional narratives translated from English into ASL. This also allows us to integrate trends towards more robust evaluation from speech and text modeling [32, 15, 10], where models trained on large web corpora are evaluated both zero-shot and finetuned on independently constructed benchmarks.

Practices for constructing test sets in prior sign language dataset works are mixed. For example, OpenASL [39] and AfriSign [17] construct their test sets by randomly splitting at the sentence level; SP-10 [42] does the same but with multiway translations identified as a single sentence. How2Sign [13] samples document-level narratives rather than individual sentences, but most signers are shared between the train and test sets, and some narratives are present in both the train and test sets, translated by different signers. BOBSL [2] invests substantial effort into creating signer-independent, topic-balanced splits; this is perhaps why its translation baseline scores only 1.00 BLEU despite the dataset's size. Zero-shot evaluation lets us sidestep these issues and get a better sense of the model's quality for real use.

Table 3: Metrics for ASL to English translation on How2Sign. Our models either train from scratch or finetune a pretrained T5 checkpoint, and are trained on How2Sign (H2S) only, YouTube-ASL (YT-ASL) only, a mixture of H2S and YT-ASL, or YT-ASL and then finetuned on H2S.

| Approach | Training Schedule | BLEU-1 | BLEU-2 | BLEU-3 | BLEU | BLEURT |
|---|---|---|---|---|---|---|
| Álvarez et al. [3] | H2S | 17.40 | 7.69 | 3.97 | 2.21 | - |
| GloFE-VN [25] | H2S | 14.94 | 7.27 | 3.93 | 2.24 | 31.65 |
| Tarrés et al.[40] | H2S | 34.01 | 19.30 | 12.18 | 8.03 | - |
| Ours (no pretraining) | H2S | 13.92 | 4.69 | 1.82 | 0.86 | 30.65 |
| | YT-ASL | 14.53 | 5.47 | 2.61 | 1.41 | 29.55 |
| | YT-ASL + H2S | 28.60 | 14.56 | 8.68 | 5.60 | 37.72 |
| | YT-ASL → H2S | 28.38 | 15.41 | 9.55 | 6.26 | 39.40 |
| Ours (pretrained) | H2S | 14.96 | 5.11 | 2.26 | 1.22 | 29.98 |
| | YT-ASL | 20.93 | 10.35 | 6.14 | 3.95 | 34.98 |
| | YT-ASL + H2S | 36.35 | 23.00 | 16.13 | 11.89 | 44.78 |
| | YT-ASL → H2S | **37.82** | **24.13** | **16.92** | **12.39** | **46.63** |

## 5.1 Setup

We ablate across four different training schedules:

- **H2S**. We train only on How2Sign, not YouTube-ASL, for a like-for-like comparison with prior methods.
- **YT-ASL**: We train only on YouTube-ASL, and evaluate on How2Sign zero-shot.
- **YT-ASL + H2S**: We train on a mixture of How2Sign and YouTube-ASL, mixed in proportion to the size of the datasets.
- **YT-ASL → H2S**: We train on YouTube-ASL, then finetune on How2Sign.

We also ablate the effect of pretraining on English text by comparing models trained from scratch using the T5.1.1-Base architecture, vs. finetuned from the T5.1.1-Base pretrained checkpoint.

We train with a batch size of 128 and learning rate of 0.001 with Adafactor [36]; other hyperparameters are the T5X defaults. For models trained solely on How2Sign data, we train for 20,000 steps. For models trained on YouTube-ASL (including with How2Sign mixed in), we train for 200,000 steps. When finetuning on How2Sign after training on YouTube-ASL, we finetune for an additional 5,000 steps. Each 1,000 steps takes approximately 0.25 TPUv4-hours.

Following prior work, we present BLEU [29] and BLEURT [34] scores. BLEU scores are computed using SacreBLEU [30] version 2, with all default options. BLEURT scores are computed using checkpoint BLEURT-20 [31, 14]. We decode using beam search with a beam width of 5.

## 5.2 Results

See Table 3 for metrics comparing our models to prior works on How2Sign [3, 25, 40]. The best results come from training on YouTube-ASL from a pretrained checkpoint, then finetuning on How2Sign, which achieves 12.39 BLEU vs. the state of the art of 8.03 BLEU [40]. The base model achieves 3.95 BLEU zero-shot, which is nontrivial but substantially worse than the finetuned score. Factors that could contribute to this gap include train/test leakage of signers and narratives, How2Sign's narrow domain, and the extra ~10% training data it represents.

Results are substantially worse when training from scratch, which suggests that T5's English pretraining gives the model a better initialization, as De Coster et al. [11] found for frozen pretrained language models. Results are absymal when trained without YouTube-ASL. The most direct comparison of our approach to prior work is T5 trained from scratch on How2Sign only, which reaches just 0.86 BLEU, despite training on the same data as Tarrés et al. [40]'s 8.03 BLEU. This might be explained by their use of a pretrained video encoder and various decisions they made to optimize for small amounts of data (smaller network, more text normalization, careful hyperparameter sweep), whereas we used a less tuned configuration that was intended for larger datasets.

Table 4: Qualitative examples from our best finetuned and zero-shot models, on sentences sampled from How2Sign by Tarrés et al. [40]. See Table 5 in the Appendix for the complete set of examples.

|     |     |     |
| --- | --- | --- |
| (1) | **Reference** | And that's a great vital point technique for women's self defense. |
|     | Tarrés et al. | It's really a great point for women's self defense. |
|     | Ours (zero-shot) | It's really great, especially for women who are facing barriers. |
|     | Ours (finetuned) | It's really great for women's self defense. |
| (2) | **Reference** | In this clip I'm going to show you how to tape your cables down. |
|     | Tarrés et al. | In this clip I'm going to show you how to improve push ups. |
|     | Ours (zero-shot) | This video will show how to use the code online. |
|     | Ours (finetuned) | In this clip we're going to show you how to cut a piece of clay. |
| (3) | **Reference** | In this segment we're going to talk about how to load your still for distillation of lavender essential oil. |
|     | Tarrés et al. | Ok, in this clip, we're going to talk about how to fold the ink for the lid of the oil. |
|     | Ours (zero-shot) | This video will discuss how to submit a digital form for the survey. |
|     | Ours (finetuned) | In this clip we're going to talk about how to feed a set of baiting lizards for a lava field oil. |

See Table 4 for qualitative examples of the translations produced by our best finetuned and zero-shot models, on sentences sampled from How2Sign by Tarrés et al. [40]. The translations capture elements of the reference translation but are clearly not yet of usable quality. The zero-shot predictions hew less closely to the references, but the errors usually make sense in light of the sign language input. For example, in (1), the sign used to mean "defense" also means "barrier".

## 6   Conclusion

In this paper, we presented YouTube-ASL, a new, publicly available parallel corpus for American Sign Language and English that is ~3x the size and has ~10x as many unique signers as the largest prior ASL dataset. Our key improvement over prior work is that we used automatic tagging followed by human filtering to increase mining recall without harming precision. We demonstrated the value of this data with a simple baseline built from off-the-shelf components (MediaPipe Holistic and T5) that achieves a new finetuned state of the art in ASL to English translation on How2Sign, 12.39 BLEU. We also reported a zero-shot score of 3.95 BLEU, a first for sign language translation. We hope that YouTube-ASL will be immediately useful for research on methods for sign language translation and caption alignment, as well as tools for automatic annotation/filtering of new sign language datasets. Because YouTube-ASL has so much signer variety, including across dialect and skill level, it may be less useful for generation than recognition tasks.

While our baseline improves upon prior work, even the finetuned translations are subjectively low-quality and are not yet useful in the real world. We hope that more refined modeling approaches will provide better results with the same data, but despite our and prior efforts, ASL is still a low-resource language by modern standards [20]—let alone the many other sign languages of the world, most of which are even less resourced. Future work may look to address this by mining broader datasets with other kinds of supervision, and exploring multilingual transfer at larger scales. As model quality improves, future work should perform more comprehensive evaluations to understand differences across domains, dialects, levels of fluency, signer appearance, and other such factors.

## 7   Ethical Considerations

Sign language datasets pose privacy challenges, because the signer's appearance (body, facial expressions), which is personally identifying, is a vehicle for the language itself. Video anonymization techniques are not yet mature enough to be useful in this regard. Our corpus is composed of sign language content that uploaders made publicly visible on YouTube, and we release only video IDs so that changes to the underlying videos are automatically reflected in the corpus. While the corpus cov-

ers a broader variety of channels than prior works, this does not mean it is necessarily representative of the signing population—or even if it were representative, that models trained on it would work equally well for everyone.

We train our models on reduced poses as a form of anonymization, but this is not suitable for all modeling approaches and may harm model quality. Until sign language translation models are closer to usable quality, there is little risk of societal harm, except that individuals or organizations mistakenly rely on models that are inadequate. As we approach that point, sign language processing will adopt the risks of natural language processing in general, but with a great potential to improve accessibility for Deaf/Hard of Hearing people.

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

# A Appendix

This document contains supplementary material for the YouTube-ASL paper.

## A.1 Full Qualitative Results

Table 5: The complete set of qualitative examples from our best finetuned and zero-shot models, on sentences sampled from How2Sign by Tarrés et al. [40].

| | | |
|---|---|---|
| (1) | **Reference** | And that's a great vital point technique for women's self defense. |
| | Tarrés et al. | It's really a great point for women's self defense. |
| | Ours (zero-shot) | It's really great, especially for women who are facing barriers. |
| | Ours (finetuned) | It's really great for women's self defense. |
| (2) | **Reference** | In this clip I'm going to show you how to tape your cables down. |
| | Tarrés et al. | In this clip I'm going to show you how to improve push ups. |
| | Ours (zero-shot) | This video will show how to use the code online. |
| | Ours (finetuned) | In this clip we're going to show you how to cut a piece of clay. |
| (3) | **Reference** | In this segment we're going to talk about how to load your still for distillation of lavender essential oil. |
| | Tarrés et al. | Ok, in this clip, we're going to talk about how to fold the ink for the lid of the oil. |
| | Ours (zero-shot) | This video will discuss how to submit a digital form for the survey. |
| | Ours (finetuned) | In this clip we're going to talk about how to feed a set of baiting lizards for a lava field oil. |
| (4) | **Reference** | You are dancing, and now you are going to need the veil and you are going to just grab the veil as far as possible. |
| | Tarrés et al. | So, once you're belly dancing, once you've got to have the strap, you're going to need to grab the thumb, and try to avoid it. |
| | Ours (zero-shot) | he's dancing a lot. Now he needs a hat and a chain |
| | Ours (finetuned) | Their hopping and dancing is now, they're going to need their squat and squat and they're going to be able to move independently. |
| (5) | **Reference** | But if you have to setup a new campfire, there's two ways to do it in a very low impact; one is with a mound fire, which we should in the campfire segment earlier and the other way to setup a low impact campfire is to have a fire pan, which is just a steel pan like the top of a trash can. |
| | Tarrés et al. | And other thing I'm going to talk to you is a little bit more space, a space that's what it's going to do, it's kind of a quick, and then I don't want to take a spray skirt off, and then I don't want it to take it to the top of it. |
| | Ours (zero-shot) | But if you have to set up a new campfire, you have to set up a campfire. You have to do it in a campfire, or set up a tentfire. |
| | Ours (finetuned) | But if you have to set up a new campfire, there are two ways to do a low impact fire, one is a cone fire, which we have to do in the tent earlier, and the other one is to set up a campfire in a fire pan. |
| (6) | **Reference** | So, this is a very important part of the process. |
| | Tarrés et al. | It's a very important part of the process. |
| | Ours (zero-shot) | Wash your hands. |
| | Ours (finetuned) | Alright, let's get started. |

# B Datasheets for Datasets

We provide documentation of the dataset based on Datasheets for Datasets [7].

---

[7] https://arxiv.org/pdf/1803.09010.pdf

### B.1 Motivation

**For what purpose was the dataset created?** The dataset was created primarily to serve as training data for ASL to English machine translation; prior datasets are smaller and have fewer unique signers. We used human annotators to identify high-quality ASL videos with well-aligned captions, but it was not feasible to manually correct or align any of the included captions. This is generally sufficient for translation, but slightly less ideal for tasks like ASL to English caption alignment, where consistent alignment standards might be desired. The dataset is probably also less suitable for English to ASL translation, due to the signing variation across videos, though this may be addressed with methods for improved controllability.

**Who created the dataset (e.g., which team, research group) and on behalf of which entity (e.g., company, institution, organization)?** This dataset was created by Dave Uthus, Garrett Tanzer and Manfred Georg for Google.

**Who funded the creation of the dataset?** Google.

### B.2 Composition

**What do the instances that comprise the dataset represent (e.g., documents, photos, people, countries)?** Each instance is the video id of a YouTube video.

**How many instances are there in total (of each type, if appropriate)?** There are 11,093 video ids.

**Does the dataset contain all possible instances or is it a sample (not necessarily random) of instances from a larger set?** This contains a subset of videos available on YouTube of people signing in American Sign Language with English captions. This is not strictly representative of the larger set, as we have applied a combination of automatic and manual filtering techniques to find high quality videos with high-quality captions.

**What data does each instance consist of?** Each instance consists of a single video id, which represents an ASL video with associated English captions.

**Is there a label or target associated with each instance?** The English captions may be considered the target for each instance, but this depends on the task being attempted.

**Is any information missing from individual instances?** No.

**Are relationships between individual instances made explicit (e.g., users' movie ratings, social network links)?** No.

**Are there recommended data splits (e.g., training, development/validation, testing)?** No. YouTube datasets can change over time due to the nature of the platform (videos can be made private or deleted), thus there are no recommended splits.

**Are there any errors, sources of noise, or redundancies in the dataset?** Yes. During the annotation process, we allowed the annotators to mark whole channels with the same annotations, so there may be some videos which are not of the same quality as the rest of the channel. There may also be minor errors in the videos or captions that the annotators explicitly deemed acceptable.

**Is the dataset self-contained, or does it link to or otherwise rely on external resources (e.g., websites, tweets, other datasets)?** The dataset is not self-contained, as it consists of YouTube video ids only. As such, there is no guarantee that the dataset will remain constant over time. The actual videos themselves fall under YouTube's Terms of Service https://www.youtube.com/static?template=terms.

**Does the dataset contain data that might be considered confidential (e.g., data that is protected by legal privilege or by doctor– patient confidentiality, data that includes the content of individuals' non-public communications)?** No, the dataset consists of video ids for public videos only, and does not rehost any of the underlying data.

**Does the dataset contain data that, if viewed directly, might be offensive, insulting, threatening, or might otherwise cause anxiety?** The video ids comprising the dataset itself are random identifiers. The videos referenced by our video ids are hosted by YouTube and therefore subject to YouTube's community guidelines. We or our annotators did not encounter any such content.

**Does the dataset identify any subpopulations (e.g., by age, gender)?** The dataset does not identify any subpopulations.

**Is it possible to identify individuals (i.e., one or more natural persons), either directly or indirectly (i.e., in combination with other data) from the dataset?** The video ids comprising the dataset itself are random identifiers. The videos referenced by our video ids include people signing in ASL, which inherently includes the person's appearance. Our dataset does not provide any extra information about these people that was not already publicly available from their uploaded videos, and respects when videos are deleted/made private by virtue of using video ids.

**Does the dataset contain data that might be considered sensitive in any way (e.g., data that reveals race or ethnic origins, sexual orientations, religious beliefs, political opinions or union memberships, or locations; financial or health data; biometric or genetic data; forms of government identification, such as social security numbers; criminal history)?** As with the previous question, the videos referenced by our video ids may contain information about many topics, if the signer chose to discuss that information in their publicly uploaded video. Our dataset does not provide any extra information and respects deletions.

### B.3 Collection Process

**How was the data associated with each instance acquired? Was the data directly observable (e.g., raw text, movie ratings), reported by subjects (e.g., survey responses), or indirectly inferred/derived from other data (e.g., part-of-speech tags, model-based guesses for age or language)?** The videos referenced by each video id instance consist solely of data that is directly observable (uploaded videos and captions). The selection of video ids is implicitly decided by a combination of automatic and manual filtering in order to ensure a relatively high quality level.

**What mechanisms or procedures were used to collect the data (e.g., hardware apparatuses or sensors, manual human curation, software programs, software APIs)?** A combination of software programs and manual annotations were used to select preexisting YouTube videos.

**If the dataset is a sample from a larger set, what was the sampling strategy (e.g., deterministic, probabilistic with specific sampling probabilities)?** Not applicable.

**Who was involved in the data collection process (e.g., students, crowdworkers, contractors) and how were they compensated (e.g., how much were crowdworkers paid)?** The authors collected the initial collection of video ids, and annotators were hired to help annotate the videos for filtering purposes.

**Over what timeframe was the data collected?** We collected videos up to January 2022.

**Were any ethical review processes conducted (e.g., by an institutional review board)?** No.

**Did you collect the data from the individuals in question directly, or obtain it via third parties or other sources (e.g., websites)?** The dataset consists of references to videos that include people, but doesn't collect or release any new information about those people.

**Were the individuals in question notified about the data collection?** No, as we only provide video ids and no further information about the videos.

**Did the individuals in question consent to the collection and use of their data?** Not applicable.

**If consent was obtained, were the consenting individuals provided with a mechanism to revoke their consent in the future or for certain uses?** As we only provide ids and not the raw content, users removing the video will make them no longer available for use in our dataset.

**Has an analysis of the potential impact of the dataset and its use on data subjects (e.g., a data protection impact analysis) been conducted?** No.

### B.4 Preprocessing/cleaning/labeling

**Was any preprocessing/cleaning/labeling of the data done (e.g., discretization or bucketing, tokenization, part-of-speech tagging, SIFT feature extraction, removal of instances, processing of missing values)?** Yes, we filtered out videos that were not relevant, of poor quality, had poor captions, etc. The result is a list of video ids, which point to unmodified videos.

**Was the "raw" data saved in addition to the preprocessed/cleaned/labeled data (e.g., to support unanticipated future uses)?** The original set of video ids will not be made available.

**Is the software that was used to preprocess/clean/label the data available?** No.

### B.5 Uses

**Has the dataset been used for any tasks already?** None prior to the baselines provided in this paper.

**Is there a repository that links to any or all papers or systems that use the dataset?** No.

**What (other) tasks could the dataset be used for?** In addition to the intended task of ASL to English translation, the dataset could be used for related sign language understanding tasks like caption alignment, and potentially for sign language generation tasks like English to ASL translation, or sign language tasks that do not require captions.

**Is there anything about the composition of the dataset or the way it was collected and preprocessed/cleaned/labeled that might impact future uses?** The dataset was filtered for high-quality videos and captions, which includes a variety of signing styles and skill levels, as long as they can be understood by an ASL user and are captioned correctly. This amount of variety may be unideal for tasks like generation where consistency is preferred. Even though it is varied, it is still not necessarily representative of the signing community as a whole, and should be treated as such.

**Are there tasks for which the dataset should not be used?** This dataset should not be used as a benchmark for comparing models across time, because the data that YouTube-ASL is derived from will change over time with video deletions or other modifications.

**Any other comments?**

### B.6 Distribution

**Will the dataset be distributed to third parties outside of the entity (e.g., company, institution, organization) on behalf of which the dataset was created?** The dataset is open sourced.

**How will the dataset will be distributed (e.g., tarball on website, API, GitHub)?** GitHub.

**When will the dataset be distributed?** It is currently available.

**Will the dataset be distributed under a copyright or other intellectual property (IP) license, and/or under applicable terms of use (ToU)?** We release the YouTube-ASL video ids under CC BY 4.0 International license, while the actual videos/captions on YouTube are preexisting and subject to the YouTube Terms of Service (https://www.youtube.com/static?template=terms).

**Have any third parties imposed IP-based or other restrictions on the data associated with the instances?** See above for license information.

**Do any export controls or other regulatory restrictions apply to the dataset or to individual instances?** No.

### B.7 Maintenance

**Who will be supporting/hosting/maintaining the dataset?** The authors will be responsible for maintaining the dataset.

**How can the owner/curator/manager of the dataset be contacted (e.g., email address)?** By contacting any of the authors listed on the publication.

**Is there an erratum?** No.

**Will the dataset be updated (e.g., to correct labeling errors, add new instances, delete instances)?** It may be updated, and if so, updates will be communicated via the associated GitHub page.

**If the dataset relates to people, are there applicable limits on the retention of the data associated with the instances (e.g., were the individuals in question told that their data would be**

**retained for a fixed period of time and then deleted)?** Our dataset is constructed similarly to past YouTube-related datasets, in that we only provide video ids. Thus, if a user makes their YouTube video private or deletes it, this will then no longer be available for use.

**Will older versions of the dataset continue to be supported/hosted/maintained?** No, if we need to remove older versions, these will be communicated on the associated GitHub page.

**If others want to extend/augment/build on/contribute to the dataset, is there a mechanism for them to do so?** No, we are currently not planning to allow formal contributions to the dataset at this time, but others may extend the dataset on their own in accordance with the license.

## C  Additional Information

URL to the data: [https://github.com/google-research/google-research/tree/master/youtube_asl](https://github.com/google-research/google-research/tree/master/youtube_asl)

Hosting and maintenance: The data website is on GitHub under Google Research's shared GitHub repository, while the data itself is hosted on Google Research's shared Google Cloud Service.

## D  Author Statement

The authors bear all responsibility in case of violation of rights, and confirm that this dataset is open-sourced under the CC BY 4.0 International license.

