# OpenReview forum: "YouTube-ASL: A Large-Scale, Open-Domain American Sign Language-English Parallel Corpus"
_NeurIPS.cc/2023/Track/Datasets_and_Benchmarks — NeurIPS 2023 Datasets and Benchmarks Poster_

### Official Review · Reviewer_t1a9 · 2023-07-16
**Review for paper 951**

**Rating:** 9
**Confidence:** 4
**Correctness:** The dataset is constructed in a sound…
**Clarity:** The paper is well-written and easy to…

**Strengths:**

The dataset is quite large, for which the number of singers is 2519, at least 10x as the existing largest datatset, and the scope of vocabularies as well as the length of hours all reach the largest values.
The authors discusses the properties the dataset in sufficient details.
The evaluation is conducted in a reasonable way, which is compared on the test data from How2Sign. By this way, the authors also compared the pretraining mode and zero-shot mode, showing the generalization ability.


**Additional Feedback:**

The dataset is of importance and can help to advance the ALR research.


**Documentation:**

The paper provides sufficient details on data collection, annotation process, and the data properties. It also gives the URL to the dataset.

**Limitations:**

The authors discussed the limitations and future wok in the conclusion.

**Opportunities For Improvement:**

The authors discussed the limitations and future wok in the conclusion.

**Relation To Prior Work:**

The paper clearly discussed the differences against previous works.

**Summary And Contributions:**

This paper introduces a large-scale American Sign Language translation dataset, which contains more than 2500 signers, about 1000 hours of videos. It conducts evaluation on How2Sign dataset, which achieves an improved precision than SOTA methods that trained directly on How2Sign. The paper also compare the modes of using pretraining and non-pretraining.

---

> ### Author Response · Authors · 2023-08-09
>
> Thank you for the positive comments!

---

### Official Review · Reviewer_3tD6 · 2023-07-16
**A largescale ASL dataset**

**Rating:** 5
**Confidence:** 5
**Clarity:** Paper is well written and well organi…

**Strengths:**

1. Authors create the largest ASL dataset with largest number of unique signers.
2. Authors perform baseline experiments to show that having a larger dataset helps to improve the results.

**Additional Feedback:**

Please check the comments above.

**Correctness:**

Claims made by authors are mostly proved via experiments. Dataset creation and design experiments look reasonable.

**Documentation:**

Dataset is described well and dataset creation process is described in detail.

Dataset license is not clear and also since authors have not taken permission from content creator, it may lead to copyright issues.

**Ethics:**

It is not clear if the authors took the permission from content creator of the ASL YouTube video, there can be copyright issues.
It is also not clear if human annotators were compensated adequately for their work.

**Limitations:**

1. The dataset is created from YouTube and authors provide the links to the videos and do not release the data directly. But since YouTube videos can change and disappear, how is this dataset useful? In the future if someone wants to use this data, and reproduce results, how would they do that?
2. It is not clear if the authors took the permission from content creator of the ASL YouTube video, there can be copyright issues.
3. A more thorough error analysis of the model is missing. BLEU scores are not that high, it would be interesting to see why is that the case and how it could be addressed.

**Opportunities For Improvement:**

1. Authors propose a very trivial model for ASL-English translation; however recently, many SOTA models have been proposed, authors should experiment with those architectures as well.
2. Alignment of video and text is an open problem sign language research. Authors state that for segmenting videos into individual sentences, they clipped it based on the caption but this can sometimes be inaccurate, specially given that a sign can have a longer duration than the English caption. Did authors manually verify this with human annotators, if the clipped video and corresponding English sentence were aligned?

**Relation To Prior Work:**

Authors should include prior work on other sign language datasets as well. Also, authors should discuss other SOTA models for sign language translation.

1. Necati Cihan Camgoz, Oscar Koller, Simon Hadfield, and Richard Bowden. 2020. Sign language transformers: Joint end-to-end sign language recognition and translation. In Proceedings of the IEEE/CVF conference on computer vision and pattern recognition
2. Abhinav Joshi, Ashwani Bhat, Pradeep S, Priya Gole, Shashwat Gupta, Shreyansh Agarwal, and Ashutosh Modi. 2022. CISLR: Corpus for Indian Sign Language recognition. In Proceedings of the 2022 Conference on Empirical Methods in Natural Language Processing (EMNLP).
3. Abhinav Joshi, Susmit Agrawal, Ashutosh Modi, 2023, ISLTranslate: Dataset for Translating Indian Sign Language, Association of Computational Lingusitics (ACL).
4. Prem Selvaraj, Gokul Nc, Pratyush Kumar, and Mitesh Khapra. 2022. OpenHands: Making sign language recognition accessible with pose-based pretrained models across languages, Association of Computational Lingusitics (ACL).
5. Kayo Yin, Amit Moryossef, Julie Hochgesang, Yoav Goldberg, and Malihe Alikhani. 2021. Including signed languages in natural language processing. Association of Computational Lingusitics (ACL).
6. Kayo Yin and Jesse Read. 2020b. Better sign language translation with stmc-transformer. In Proceedings of the 28th International Conference on Computational Linguistics.
7. Chenchen Xu, Dongxu Li, Hongdong Li, Hanna Suominen, and Ben Swift. 2022. Automatic gloss dictionary for sign language learners, Association of Computational Lingusitics (ACL).


**Summary And Contributions:**

This paper introduces a new translation dataset for American Sign Language (ASL). The dataset has about 1000 hours of video, about 2500 signers and is thrice the previous largest dataset. The dataset is created by scraping videos from YouTube. The scrapped data is firstly, filtered for videos having captions and in the second step human annotators filter out poor quality videos.

---

> ### Author Response · Authors · 2023-08-09
>
> Thank you for your review and comments.
>
> Re Opportunities for Improvement:
>
> 1:
> We would prefer to characterize the approach as “simple” rather than “very trivial”. Even so, we are leveraging the large amount of work that has gone into MediaPipe and T5, neither of which are trivial. More importantly, leveraging pretrained foundation models has become an important area of research as models scale and become more general-purpose.
>
> While other architectures developed specifically for sign language translation would probably also benefit from this larger amount of data, we consider that work out of scope, as this is primarily a dataset paper submitted to a dataset venue. It takes a lot of work and computational resources to reproduce a variety of modeling approaches, and the results would not enhance the main contributions of our work—namely the dataset and the new methodology for constructing it.
>
>
> 2:
> We will add discussion in Section 3.2 on what it means for captions to be aligned in this context. (i.e., The captions start and stop in close proximity to the signing, and the timings appear to have been annotated based on the video rather than a voiceover. Within a sentence (which may consist of multiple captions), there may or may not be an alignment because the syntax of ASL is different from English.) But more generally, we think that we give a complete picture of what YouTube-ASL is and what limitations it has, i.e. that it is a corpus with generally good but not perfect captions that can be preprocessed/used in a variety of ways. (How to best preprocess this kind of data is itself a research question.)
>
>
>
> Re Limitations:
>
> 1+2:
> We understand that a train-only dataset consisting of links to content that might be deleted is not perfectly suited to classic machine learning modeling comparisons.
>
> But in some sense this is inherent to mining this kind of data from the web. OpenASL, being constructed from YouTube as well, has similar limitations. The OpenHands datasets you referenced are in the same boat.
>
> Outside of sign language, there are many non-text datasets that are structured as links to data hosted elsewhere (which may be deleted over time). For example, both of NeurIPS Datasets & Benchmarks 2022’s Best Papers (LAION 5B and MineDojo) released image/video data in this format.
>
>
> Given that YouTube is probably the largest repository of sign language videos on the internet, this seems well-worth the tradeoff. Compared to the largest “stable” ASL dataset, How2Sign, YouTube-ASL is >12x the size and has >200x as many unique signers; there are also more intangible qualities that come from web-mined datasets, like the diverse recording environments , varied domain , more varied level of signing proficiency, and presence of originally signed rather than translated content. We highlight these factors in the Related Work section.
>
> When testing new modeling approaches, you can also get a like-for-like comparison by training both models on your local copy of the data at the same instant in time. This isn’t ideal for comparison across papers, but you can at least get fair comparisons within a paper.
>
>
> 3:
> We think that the BLEU scores are pretty high, given that they’re 50% higher than the prior state of the art! Sign language translation is a really challenging task, and as we mention in the conclusion, even with this larger dataset ASL is still a low-resource language even by the standards of text-to-text translation.
>
> In terms of the BLEU for our data-matched How2Sign baseline: we briefly discuss in the paper why our model might perform worse than prior SOTA when trained solely on How2Sign. The most likely reason is that they use a pretrained video encoder, whereas our model has to learn to use skeletons from scratch. But Tarrés et al. (which is a modeling paper, not a dataset paper) also does a lot of optimizations specific to small datasets, whereas we use a simple approach that should be more amenable to scaling. We think that our ablations make it clear that training the same model with additional data from YouTube-ASL helps, which validates the dataset contributions of our paper.

---

> > ### Author Response · Authors · 2023-08-09
> >
> > Re Related Work:
> >
> > Could you clarify your comments? We do discuss prior sign language datasets and methods at length. Even if you are suggesting additional works for us to cite, we are still a little confused.
> >
> > Of the 7 papers that you suggest we cite (if we are understanding correctly), we already do cite 3 of them (Camgöz, Yin, Yin).
> >
> > And one of the papers you mention, “ISLTranslate: Dataset for Translating Indian Sign Language”, is indeed relevant, but it was released on arXiv/ACL several weeks after the NeurIPS Datasets & Benchmarks submission deadline, so it impossible for us to have cited it.
> >
> > For the remaining papers:
> > Thank you for suggesting the OpenHands unlabelled ISL dataset and CISLR isolated sign classification dataset. We will mention them (and the more recent work in OpenHands “Addressing Resource Scarcity across Sign Languages with Multilingual Pretraining and Unified-Vocabulary Datasets”) at the end of Section 2.1, along with the other web-mined datasets for non-translation tasks. They are very similar in approach to OpenASL in that they searched for the channels manually, so our two step automatic tagging+human filtering approach is still novel for sign language.
> >
> > Xu does not seem relevant, as it is an isolated sign modeling paper on top of the WLASL dataset (which we do cite) and doesn’t seem to originate any new pose modeling techniques.
> >
> >
> >
> > Re Documentation and Ethics:
> >
> > In the supplementary Datasheet, we go into more detail on the license. We release the YouTube video IDs under CC-BY 4.0, but the videos and captions referenced by the IDs have various statuses depending on the uploader in accordance with the YouTube Terms of Service. As explained above, this is common for web-mined image/video datasets (e.g.,  YouTube-8M and Kinetics (whose I3D model is used frequently in sign language research)).
> >
> >
> > In the submission checklist, we specified that we hired a small number of contractors (the annotators) at market rates.

---

### Official Review · Reviewer_HbjK · 2023-07-21
**Reviews of HbjK**

**Rating:** 5
**Confidence:** 5
**Correctness:** The claims are correct.
**Clarity:** The paper is well written.

**Strengths:**

1. The dataset is much larger than existing ASL datasets, which is definitely helpful to the research on sign language translation.
2. The collection process is simple and cheap, which can be easily migrated to other sign languages.
3. Zero-shot experiments over How2Sign are conducted.

**Additional Feedback:**

N/A

**Documentation:**

A documentation is provided.

**Ethics:**

No ethical concerns.

**Limitations:**

The authors have adequately discussed limitations.

**Opportunities For Improvement:**

My major concerns lie in the dataset accessibility and train/test splits.

1. The dataset only comprises video IDs on YouTube. Some videos may be deleted from the platform over time, which make the dataset not "stable". It would become a major obstacle for future researchers to choose to use this dataset.
2. The authors don't provide train/dev/test splits. It will be difficult for future works to conduct a fair comparison on this dataset.
3. Only pose-based baseline results are provided. The authors may consider adding RGB-based or dual-modality (RGB+Pose) methods to enrich baseline results.

**Relation To Prior Work:**

Yes. A comparison is available in Table 1.

**Summary And Contributions:**

The paper contributes a large-scale American sign language (ASL) dataset, YouTube-ASL, which is collected from YouTube programs. The dataset is quite large with ~60K unique signs and the signers in the dataset are diverse (more than 2K signers). The dataset collection process includes two parts. First, candidate sign videos are automatically retrieved according to video tags. Second, three native signers are recruited to identify high-quality videos. Finally, several pose-based benchmark results are provided over How2Sign.

---

> ### Author Response · Authors · 2023-08-09
>
> Thank you for your review and comments.
>
> It sounds like your main concern is with respect to the dataset’s stability over time. We understand that a train-only dataset consisting of links to content that might be deleted is not perfectly suited to classic machine learning modeling comparisons.
>
> But in some sense this is inherent to mining this kind of data from the web. OpenASL, being constructed from YouTube as well, has similar limitations. Outside of sign language, there are many non-text datasets that are structured as links to data hosted elsewhere (which may be deleted over time). For example, both of NeurIPS Datasets & Benchmarks 2022’s Best Papers (LAION 5B and MineDojo) released image/video data in this format.
>
> Even for datasets that are “stable”, like C4 and other text datasets, the particular splits are not especially important because they’re just used to measure overfitting when pretraining for multiple epochs, and independent benchmarks are preferred for evaluation.
>
> Given that YouTube is probably the largest repository of sign language videos on the internet, this seems well-worth the tradeoff. While ideally, you could create a large dataset that is stable over time to allow for perfectly fair comparisons, it’s not clear to us that this is feasible. Compared to the largest “stable” ASL dataset, How2Sign, YouTube-ASL is >12x the size and has >200x as many unique signers; there are also more intangible qualities that come from web-mined datasets, like the diverse recording environments (How2Sign was recorded in a lab), varied domain (How2Sign is only “how to” instructional narratives), more varied level of signing proficiency, and presence of originally signed rather than translated content. We highlight these factors in the Related Work section.
>
> For this reason, we say in the paper that YouTube-ASL should only be used as a training dataset, not a benchmark. Instead we measure finetuned and zero-shot performance on How2Sign, which as you indicate in your Strengths section, is a contribution in itself: using a separate benchmark is a more robust form of evaluation than if we had used a test split from the same domain. In the future, we hope to create benchmarks that are more suited for evaluation purposes.
>
> When testing new modeling approaches, you can also get a like-for-like comparison by training both models on your local copy of the data at the same instant in time. This isn’t ideal for comparison across papers, but you can at least get fair comparisons within a paper.
>
>
> Re RGB models, we agree that this is an area that should be explored more in future work. We mention it as another line of modeling approaches throughout our paper, but setting up the pipeline for it is a nontrivial amount of work, on top of the resources to train another set of models and the fact that it carries a different set of privacy considerations. We think that our existing experiments demonstrate the value of YouTube-ASL as a dataset in a clear and focused way. This is ultimately a dataset paper submitted to a dataset venue, not a modeling paper, and doing this comparison right we consider to be out of scope.

---

> > ### Comment · Reviewer_HbjK · 2023-08-29
> >
> > Thanks for the authors' rebuttal. But I still think that the dataset's instability is a weakness.

---

### Official Review · Reviewer_f1hs · 2023-07-23
**Good paper, clearly written, and follows the guideline**

**Rating:** 7
**Confidence:** 3
**Correctness:** Yes, it sounds correct.

**Strengths:**

The paper is well-written, the limitations of the work is clearly explained, and justified, the rationale for the choices are presented.

**Additional Feedback:**

I provided my comments above.

**Clarity:**

The paper is well-written. I specially enjoyed that the authors at each step they explained the limitations of their work, and rationale for choices.

**Documentation:**

Yes

**Ethics:**

The authors address this in the Ethical Consideration section.

**Limitations:**

Yes, the paper has a section on ethical consideration and societal impact.

**Opportunities For Improvement:**

Some choices in the data filtering seems heuristic and not well-rationalized. For example, the authors used a heuristic approach and removed videos with durations less than 10 seconds or more than 5 hours. It would have been better if the authors justified this automatic filtering based on the nature of the ASL data. For example, what is the shortest length the video with a meaningful ASL, etc. The authors mentioned that they excluded videos which their captions are poorly aligned. But they do not explain how they evaluate the misalignment.


**Relation To Prior Work:**

Yes

**Summary And Contributions:**

The authors present a large scale American Sign Language dataset. The dataset is drawn from YouTube channels for sign language and then evaluating them by three professional annotators.

---

> ### Author Response · Authors · 2023-08-09
>
> Thank you for your review and comments.
>
> We will expand in Section 3.2 in the following ways:
>
> We will clarify that removing videos under 10 seconds doesn’t have a substantial impact on sign language videos with captions, because videos under 10 seconds (like the isolated sign videos mentioned in the paper) tend to rely on the video title rather than an actual caption track.
>
> We will explain why we excluded videos over 5 hours, namely, that from inspection they were either live interpreted broadcasts (which did not have aligned captions) or irrelevant content (corrupt, or mostly static content for hours on end).
>
> We will explain what it means for captions to be well-aligned in this context. (i.e., The captions start and stop in close proximity to the signing, and the timings appear to have been annotated based on the video rather than a voiceover. Within a sentence (which may consist of multiple captions), there may be but is not necessarily an alignment because the syntax of ASL is different from English.)
>
> We ultimately agree that some of the steps are heuristic: we call them that in the paper. We arrived at the heuristics by iteratively mining and auditing the resulting data, trying to find the right balance of filtering out useless data while retaining enough relevant data. We picked relatively conservative values, so that very little useful data was dropped (i.e., sample statistics wouldn’t be very meaningful because the rate of useful data lost was so low). Annotating the videos was a time-consuming process, and it was especially important to filter out videos using these heuristics as a first pass because we wanted to reduce the amount of annotator time wasted looking at irrelevant videos.

---

### Official Review · Reviewer_Px9y · 2023-07-23
**Great Corpus!**

**Rating:** 6
**Confidence:** 4
**Correctness:** Yes
**Clarity:** Yes

**Strengths:**

1. This dataset is a timely and high-quality contribution to an under-investigated area with significant societal value.
2. The improvement in model performance from a BLEU score of 1.22 to 12.39 is non-trivial and meaningful.
3. The dataset has the potential to further advance the field by making Sign Language Translation (SLT) a real-world application. It provides a valuable foundation for future research in this area.

**Additional Feedback:**

Please discuss "Opportunities For Improvement" and "Documentation" parts in the rebuttal.

**Documentation:**

No. **Could the authors provide the code and scripts to reproduce the results, especially the last row of the main table, i.e., YT-ASL → H2S 37.82 24.13 16.92 12.39 46.63**？

**Ethics:**

 The authors have discussed this in the paper. I also agree with the authors' assertion that until sign language translation models are closer to usable quality, there is little risk of societal harm.

**Opportunities For Improvement:**

1. The overall construction of the dataset is somewhat simplistic. It would be beneficial if the authors could elaborate on the design principles used in creating the dataset, which could contribute to the construction of similar datasets for other languages.

2. Another valuable point for consideration is whether the ASL-English model could serve as a parent model for transfer learning to ASL-German or ASL-Chinese, thereby improving the performance of other low-resource SLT tasks. If effective, this would significantly enhance the value of this dataset. For example, the ASL-German/ASL-Chinese could refer to the work presented in https://openreview.net/pdf?id=EBS4C77p_5S, ICLR 2023. **I would like to increase my score from 6 to 7/8 if the authors could discuss this point in the rebuttal.**

**Relation To Prior Work:**

Yes

**Summary And Contributions:**

The paper discusses the creation of a new corpus for sign language translation (SLT), namely YouTube-ASL, which consists of 11,093 American Sign Language  (ASL) to English videos with 984 total hours of footage. The researchers used automatic tagging and human filtering to label the data. They demonstrated the value of this data with a simple baseline built from off-the-shelf components (MediaPipe Holistic and T5) that achieves a new fine-tuned state of the art in ASL to English translation on How2Sign, with a BLEU score of 12.39.

---

> ### Author Response · Authors · 2023-08-09
>
> Thank you for your review and comments.
>
>
> Regarding 1. We were hoping you could clarify what part of the dataset construction design principles you would like us to elaborate more on. As we discussed in the paper, we wanted to scale up to more data without prohibitive annotation costs, so we decided to use annotators to find videos that were already relatively high quality, rather than fix up the videos themselves. This allowed us to mine videos from more content creators (i.e. get more unique signers), which makes the dataset more relevant for sign language understanding tasks and less immediately applicable to sign language generation tasks. We explained how we automatically mined candidate videos on YouTube, including what heuristics we used for filtering. These heuristics were created through an iterative process to help balance out what videos we wanted to include as part of the annotation process (videos that would have enough content to be usable and included uploaded captions) and what videos we wanted to exclude (often live interpreted broadcasts or corrupt videos), to reduce annotator waste. These filtered videos were then sent to human annotators to filter out unacceptable videos, with the acceptance criteria described in Section 3.2.
>
>
> Regarding 2. It’s a great point that this data could be useful for multilingual and multitask transfer during pretraining. We already reference SLTUNet (and other similar works like AfriSign) in related work, saying “Other works seek to benefit from transfer from spoken language or other sign language data”, but we agree that it should be emphasized more throughout the paper. We updated some of the contextualization in the introduction and conclusion to reflect this.
>
> We expect that sign languages will benefit from transfer like spoken languages do, and perhaps even more so, because the model has to learn representations of human body motion and some relatively universal aspects of sign language grammar (like use of space), which don’t exist for text. Experiments for this were outside the scope of this paper (as it is already a challenging task just going ASL → English, and there are some license issues that make it difficult for us to use datasets for the other languages you mention), but it is something we are planning for future work.
>
> Documentation: Could you clarify what you mean by this? In the submission checklist, we note that we can’t release our model training code because it uses an internal framework that hasn’t been open sourced. For the actual metrics, we do cite what libraries/versions we are using to compute SacreBLEU and BLEURT, along with using default options for the former. We no longer have the predictions that were scored, because they and the model checkpoint have been automatically deleted for privacy reasons.

---

### Decision · Program_Chairs · 2023-09-22

**Decision:**

Accept (Poster)

**Comment:**

This paper presents YouTube-ASL, the largest (as of now) American Sign Language to English captions dataset. The dataset is created from curating videos on YouTube and contains 1’000 hours of videos from 2’500+ unique signers.
The authors show the validity of the dataset by training a simple model on the dataset and show a 12.3 BLEU points improvement over the baseline.

There is a strong variance between reviewers, and some limitations have been noted: only the video ids are being provided (which is very common on web-crawled dataset), no code is provided to reproduce the ressource in other languages, the baseline is quite simple and more up-to-date approaches could have been used (such as adapting a pre trained model on YT-ASL).

While I agree with the limitations presented by the reviewers, this is an under-investigated area with significant societal value, which would benefit from the spotlight of the conference.

Recommendation: poster accept